# Progress on SDG 7 achieved by EU countries in relation to the target year 2030: A multidimensional indicator analysis using dynamic relative taxonomy

**Marek Walesiak[1], Grażyna Dehnel[2]***

**1** Department of Econometrics and Computer Science, Wroclaw University of Economics and Business, Wrocław, Poland, **2** Department of Statistics, Poznań University of Economics and Business, Poznań, Poland

* grazyna.dehnel@ue.poznan.pl

**Data Availability Statement:** The data underlying the results presented in the study are available from Eurostat website: "Annual EU SDG indicator

## Abstract

In 2015, 193 UN members adopted the resolution "Transforming our world: the 2030 Agenda for Sustainable Development", which set out 17 Sustainable Development Goals to be achieved by 2030. The aim of the study is to assess progress towards meeting SDG 7 "Ensure access to affordable, reliable, sustainable and modern energy for all" by individual EU countries in 2010–2021 and to determine their distance in relation to the target set for 2030. Eurostat monitors and assesses progress towards SDG 7 using seven indicators. These indicators were used to create an aggregate index. In order to limit the impact of the compensation effect on the ranking of EU countries, we applied dynamic relative taxonomy with the geometric mean to create an aggregate measure that takes into account target values for the indicators with adjusted data. The study reveals systematic progress towards reaching the EU's SDG 7 in the period 2010–2021, with differences between individual EU countries clearly decreasing. The smallest distance in relation to the target set for SDG 7 can be observed for Sweden, Denmark, Estonia, and Austria. By far the greatest progress in period 2010–2021 has been achieved by Malta, and significant for Cyprus, Latvia, Belgium, Ireland, and Poland.

## 1. Introduction

### 1.1. The characteristics of SDG 7 goal, variables, and target levels

In September 2015, 193 United Nations member states adopted the resolution "Transforming our world: the 2030 Agenda for Sustainable Development" containing 17 Sustainable Development Goals [1]. Goal 7: "Ensure access to affordable, reliable, sustainable and modern energy for all" includes three core targets, which are to be achieved by 2030 [2, p. 19/35]:

7.1.  Ensure universal access to affordable, reliable, and modern energy services.

7.2.  Increase substantially the share of renewable energy in the global energy mix.

review" (https://ec.europa.eu/eurostat/web/sdi/information-on-data).

**Funding:** The possibility of funding for publication within the 'Regional Initiative for Excellence programme of the Minister of Education and Science of Poland, years 2019-2023" (grant no. 004/RID/2018/19) ended at the end of 2023. Currently, the publication can be funded by two universities: Uniwersytet Ekonomiczny w Poznaniu (50%) and Uniwersytet Ekonomiczny we Wrocławiu (50%). The funder does not play any role in the study design, data collection and analysis, decision to publish, or preparation of the manuscript.

**Competing interests:** The authors have declared that no competing interests exist.

7.3.  Double the global rate of improvement in energy efficiency.

7.a.  Enhance international cooperation to facilitate access to clean energy research and technology, including renewable energy, energy efficiency and advanced and cleaner fossil-fuel technology, and promote investment in energy infrastructure and clean energy technology.

7.b.  Expand infrastructure and upgrade technology for supplying modern and sustainable energy services for all in developing countries, in particular least developed countries, small island developing States and landlocked developing countries, in accordance with their respective programs of support.

Progress towards reaching core targets of SDG 7 is measured by selecting an appropriate set of indicators. The global indicator framework (United Nations, 2022) contains 248 SDG Indicators for 17 SDGs. Indicators for SDG 7 included in the UN publication and 2030 targets for OECD countries based on [3] are presented in Table 1.

The main research goal of the study described in this is to assess progress in achieving core targets of SDG 7 by EU member states in the period 2010–2021 and determining their distances in relation to the goals set for 2030. Since 2017 Eurostat has published an "Annual EU SDG indicator review" (https://ec.europa.eu/eurostat/web/sdi/information-on-data), which contains an updated list of indicators for 17 SDGs (SDGs in the EU context). Seven indicators, defined in 2017 to monitor progress on SDG 7, have not changed until now. They are systematically monitored and assessed by Eurostat. Only three indicators from this list are consistent or partially consistent with UN Global list indicators (see Table 2).

The following 2030 targets for the seven indicators used to monitor progress on SDG 7 achieved by UE member states have been adopted in the study (see the last column in Table 2):

a. In the case of „Primary energy consumption", „Final energy consumption" and "Share of renewable energy in gross final energy consumption", targets for 2030 are the same as those set by the European Commission [7]. Since indicators of energy consumption (primary, final) are expressed in absolute terms (million tonnes of oil equivalent–mtoe), they cannot be used to compare different EU countries because these quantities are not directly comparable. For this reason, absolute values were replaced with indices representing changes in primary and final energy consumption in relation to values recorded in 2005 (Index 2005 = 100);

b. If no target value can be found in the 2030 Agenda, it is based on "best performance" among EU countries for 2015 year. This is defined as:

**Table 1. Indicators and targets for SDG 7.**

| Code | Indicators (UN Global list) | Unit | 2030 targets for OECD countries |
|------|------------------------------|------|----------------------------------|
| 7.1.1 | Proportion of population with access to electricity | % | 100 |
| 7.1.2 | Proportion of population with primary reliance on clean fuels and technology | % | 95 |
| 7.2.1 | Renewable energy share in the total energy consumption | % | 58.62 |
| 7.3.1 | Energy intensity measured in terms of primary energy and GDP | Megajoules per US $ | NA |
| 7.a.1 | International financial flows to developing countries in support of clean energy research and development and renewable energy production, including in hybrid systems | current US $ | NA |
| 7.b.1 | Installed renewable energy-generating capacity in developing countries | watts per capita | NA |

Source: Authors' compilation based on [2] and [3, p. 49].

**Table 2. Indicators and targets for SDG 7 for EU countries.**

| Headline indicators | Code in the UN global list | Variable type | Symbol used in study | Unit | EU-level targets for 2030 | |
|---|---|---|---|---|---|---|
| Primary energy consumption | NA | D | – | mtoe | 1023 | (!) |
| | | | y1 | 2005 = 100 | 68.30 | |
| Final energy consumption | NA | D | – | mtoe | 787 | (!) |
| | | | y2 | 2005 = 100 | 75.58 | |
| Final energy consumption in households per capita | NA | D | y3 | kgoe | 319.8 | (!!) |
| Energy productivity | 7.3.1 (s) | S | y4 | Euro per kgoe | 10.44 | (!!) |
| Share of renewable energy in gross final energy consumption | 7.2.1 (i) | S | y5 | % | 40 | (!) |
| Energy import dependency | NA | D | y6 | % | 24.58 | (!!) |
| Share of population unable to keep home adequately warm | 7.1.1 (a) | D | y7 | % | 1.88 | (!!) |

(i)–identical, (s)–similar, (a)–alternative indicator.

S–stimulants (where higher values are more preferred), D–destimulants (where lower values are more preferred).

(!)–level set by the European Commission; (!!)–the level attained in 2015 by the top 10% of EU27 countries.

Source: Authors' compilation based on data from Eurostat [4–6] and the European Commission [7].

- the 90th percentile for stimulants–i.e., the level attained in 2015 by the top 10% of EU27 countries. A similar approach was proposed by OECD [3, p. 23] taking into account data for OECD countries;

- the 10th percentile for destimulants–i.e., the level attained in 2015 by the top 10% of EU27 countries.

The analysis is based on average data for the European Union and for 27 EU countries separately from the period 2010–2021. The study includes Croatia, which joined the EU in 2013 and excluded the UK, which left the EU in January 2020. All statistical data come from Eurostat.

Of the seven indicators listed in Table 2, the biggest number of countries achieved in 2021 the EU target for x7 (Share of population unable to keep home adequately warm). The target was set at 1.88%, and the group of countries that managed to achieve this target includes Sweden, Finland, Slovenia, and Austria. The x7 indicator is strongly associated with low levels of income in combination with high expenditure on energy and poor building efficiency standards. Starting from 2012, access to affordable energy for all European Union countries systematically improved until 2019, just before the outbreak of the pandemic. Countries of Northern Europe and most of those in Western Europe had the lowest shares of people without affordable access to heating, in contrast to countries of Southern and South-Eastern Europe, which suffered from the lack of adequate heating. This was mainly due to the poor energy efficiency of buildings, the lack of adequate heating systems and insulation, which caused higher heating costs. In addition, the generally lower income levels in these regions affect housing standards and the ability to pay for fuel [8].

In the case of the x5 indicator (Share of renewable energy in gross final energy consumption), related to energy supply, the target for 2030 (40%) was achieved by Sweden, Finland, and Latvia. In 2021 the share of renewable energy sources (RES) in gross final energy consumption was equal to 62.6% in Sweden, 43.1% in Finland and 42.1% in Latvia. Such results were achieved by relying on hydropower and solid biofuels, which are regarded as more environmentally friendly compared to conventional energy sources. In general, the use of

renewable energy in the EU can be said to be steadily growing. This growth is largely due to the use of wind and solar energy. However, the variation in the share of renewables across Member States is still very large. This can be attributed, among other things, to differences in the availability of RES as well as the degree of available financial and regulatory support. Compared to 2005, the share of renewable energy sources in 2021 more than doubled, from 10.2% to 21.8% of gross final energy consumption. This increase was driven by reductions in investment costs, the use of more efficient technologies, supply chain improvements and support schemes for renewables [7]. It is worth noting that the 9.5% increase in the share of renewable energy in gross final energy consumption in 2019–2021 is also the result of a decline in final energy consumption during the COVID-19 pandemic, which means that this change was temporary. Once final energy consumption returns to pre-pandemic levels, the share of RES in gross final energy consumption will most likely fall [5].

The 2030 target for the x6 indicator (Energy import dependency), which is related to energy supply, was achieved in 2021 by Sweden and Estonia. In 2020, all EU countries were net importers of energy, with 16 importing more than half of their total energy consumption from other countries (EU and non-EU countries). Compared to 2005, fuel imports from non-EU countries slightly decreased from 57.8% to 57.5% of gross available energy. In 2020, the main non-EU energy suppliers included Russia (43.6% of gas, 28.9% of petroleum products and 53.7% of solid fuel imports), Norway and the United Kingdom (25.4% of gas imports and 16.5% of oil imports), North America (18.8% of solid fuel imports). Imports of fossil fuels still cover more than half of the EU's energy demand despite the continuous growth of renewable energy sources. The stagnation is due to two factors. First, the EU has reduced energy consumption and increased the use of domestic renewables. Second, primary production of fossil fuels has declined due to the depletion or uneconomical exploitation of domestic sources, especially in the case of natural gas [5].

In 2021, Spain, Malta and Portugal achieved the 2030 target for the x3 indicator (final energy consumption in households per capita), which is related to energy consumption. Households account for about a quarter of final energy consumption. From 2010 to 2015, household energy consumption in EU countries decreased by 12.7%, and remained at more or less the same level for the next five years: in 2020, it was only about 0.5% higher than in 2015. It was not until 2021, that the indicator had increased by 5.6% compared to 2020. Thanks to improvements in energy efficiency, particularly in space heating, it was possible to balance the population growth and increases in the number and size of dwellings [5]. In the period from 2010 to 2021, energy consumption per capita in the EU decreased by 7.3%. This decline was accompanied by a slight downward trend in total household energy consumption offsetting a 1.3% increase in the population (https://ec.europa.eu/eurostat/databrowser/view/demo_gind/default/table?lang=en).

In 2021, Denmark, Ireland and Luxembourg achieved the 2030 target for the x4 indicator (energy productivity), related to energy consumption. Since 2000, the EU continuously increased its energy productivity, reaching EUR 8.55 per kilograms of oil equivalent (kgoe) in 2021. All Member States contributed to this positive trend in an attempt to reach the 2030 target of 10.44 per kgoe.

As regards the last two indicators, x1 (primary energy consumption) and x2 (final energy consumption), related to energy consumption, in 2021 the target set for 2030 was only achieved by Greece. From 2005, primary and final energy consumption in the EU was generally on a downward trend, which was due to various factors, including a structural shift towards less energy-intensive industries and improvements in end-use efficiency in the residential sector. Measures taken during the COVID-19 pandemic and the related restrictions on public life and economic activity resulted in a significant decrease in energy consumption by

over 8% in 2020 compared to 2019. Other factors that in the long term contribute to falling energy consumption include improvements in energy efficiency and the growing use of energy from renewable sources. Despite the fact that effects of the pandemic can still be felt in the economy, the recovery observed in 2021 certainly led to higher energy consumption, preventing EU countries from achieving the 2030 target. Therefore, innovation and additional improvements in energy efficiency are still required.

## 1.2. The purpose of the article

Both the European Union as a whole and each of its member states analysed in the study vary in their progress on SDG 7. To adequately describe this problem, it is necessary to use different research approaches. This is also confirmed by the existing literature. Three methodological approaches can be distinguished in this regard:

1. The aggregate measure approach, in which one measures the overall progress on all indicators achieved by individual countries. This approach can be used to create a ranking of countries according to the value of the aggregate measure of all indicators (diagnostic features) representing progress on SDG 7. Because this approach does not take into account the target values these indicators should achieve by 2030, it cannot be used to determine the distance of individual EU countries in relation to targets set for 2030. This approach is applied, among others, by [9–13].

2. The aggregate measure approach which takes into account the target values for the indicators (see Table 2) without adjusting the data. National values of individual indicators are related to their target values. This approach can also be used to create a ranking of countries according to the value of the aggregate measure of all indicators (diagnostic features) representing progress on SDG 7. In addition, it is also possible to calculate how far individual EU countries are from achieving the goals set for 2030 (EU-level targets for 2030). This kind of approach was presented, for example, in the work by [3, 14].

3. The aggregate measure approach which takes into account the target values for the indicators (see Table 2) with adjusted data. In addition to assessing progress on SDG 7 achieved by individual EU countries and their distance in relation to the goals set for 2030, the approach relies on data corrected as follows: when a given country exceeds targets shown in Table 2 (EU-level targets for 2030), these higher national values for stimulants and lower national values for destimulants are not included in the analysis but are replaced by target values set for the entire European Union. This approach was presented, for example, in the work by [15, 16] to assess the implementation of the Europe 2020 Strategy.

The purpose of the following study is to assess progress made by individual EU countries in 2010–2021 towards achieving SDG 7 and to determine their distance in relation to targets set for 2030. To overcome the problems with the first and second approach, we propose an innovative third approach, which accounts for the target values of the indicators and uses adjusted data. This approach has not been used in this type of research so far and the use of the geometric mean in the dynamic relative taxonomy method has made it possible to reduce the impact of the negative compensation effect on the values of aggregate measures. The results of the study have important implications for individual EU countries. In addition to showing progress towards the SDG 7 target, they also represent the distance that separates each country from achieving the 2030 target. The proposed methodology can also be used to assess progress made by EU countries in the implementation of other SDGs of the 2030 Agenda.

Composite indicators play an important role in the analysis of socio-economic phenomena. A number of different approaches to constructing composite indicators have been proposed in the literature [17, p. 3] depending on the degree of compensation: compensatory, partially compensatory, and non-compensatory. Studies on multi-criteria decision-making uses of non-compensatory methods, see [18–20]. An overview of compensatory and partially compensatory aggregate measures applied to different types of data can be found in [21, 22].

Under the aggregate measure approach, positive and negative deviations from the target values of individual indicators can accumulate. As a result, countries which have exceeded targets for some indicators but have failed to achieve those set for the majority of other indicators can be classified as countries that have made much progress on SDG 7. In other words, the main weakness of methods based on aggregate measures is their compensatory nature, which is most strongly manifested in the first and the second approach.

In order to limit the impact of this compensation effect on the ranking of EU countries, the following were included in the methodology of relative taxonomy (see section 3):

- the third aggregate measure approach which takes into account the 2030 target values for the indicators (see Table 2) with data adjustment,

- in step 6 and 7 of the procedure, the geometric mean was used in the construction of the aggregate measure.

## 2. Literature review

In recent years, energy security has become a basic element of the economic security system. Recent events have clearly shown that energy policy and its monitoring play a key role in ensuring stable economic development [23, 24]. Without an energy policy, it is difficult to guarantee the security of energy supply to households and other consumers from all economic sectors. The Sustainable Development Goals (SDGs) set out by the UN in 2015 are particularly relevant in the present political and economic context [1, 25]. The 17 goals, including 169 targets, indicate global priorities which comprehensively define sustainable development in terms of economic, environmental, and social aspects [26]. SDGs are a revised version of the Millennium Development Goals (MDGs) formulated for the period 2000–2015. The operational period of the MDGs revealed that these goals did not focus enough on some issues related to sustainable development, such as energy [27, 28]. Although access to sustainable energy services is one of the fundamental conditions for sustainable development, energy was only included as a key theme in Agenda 2030 [1].

SDG 7 is to "ensure access to affordable, reliable, sustainable, and modern energy for all". This goal includes five targets to be achieved by 2030 (see section 1.1; [29]) and is closely related to the targets of eight other goals. Many studies on the quality of interlinkages between SDGs have confirmed that the pursuit of certain targets generates effects that have an impact on other targets [25, 30]. Researchers highlighted positive and negative effects [31–33]. A number of studies have been undertaken to identify correlations and map relationships between the SDGs [28, 34–40]. Some have focused only on a specific target area such as energy, water or food and explored its links with other SDGs [41–43]. SDG 7 has been found to be very strongly correlated with SDG 11 (Sustainable Cities and Communities), SDG 12 (Responsible Consumption and Production) and SDG 13 (Climate Action).

While studies on interlinkages between SDGs are helpful in evaluation processes, they do not provide complete information on whether, and to what extent, the SDGs are actually achieved. To maximize progress on SDGs, it is necessary to assess the importance of the individual goals by identifying problems and barriers to achieving them and areas that require

attention in the future [44]. Undoubtedly, one of the key questions that needs to be answered is how to measure countries' performance and evaluate their actions aimed at achieving SDGs. One of the priorities indicated by [45] for how the scientific community should participate in this process was to design a way for tracking and assessing progress on each SDG. Given the wide scope of the 17 goals, different scales (national, regional, global), multi-issue coverage and ambiguous language, such assessment requires appropriately adjusted statistical tools [10, 46, 47]. Even at the level of each country, the position in a particular hierarchy is often relative and depends on initial assumptions, the selected method, indicators used to create the ranking. Additionally, some goals (e.g., SDG 7) are more sensitive to methodological choices than others (e.g., SDG 16) [14, 48, 49].

Although the measurement of progress on the SDGs has been the topic of debate among researchers since the adoption of the 2030 Agenda, the literature concerning possible approaches is still limited [14, 46, 47, 50]. Since the SDGs and targets cannot be measured directly, they have been mostly operationalized by a number of indicators [51, 52]. 231 indicators were defined by the UN Statistical Commission to monitor and assess global sustainability [53, 54]. The set of SDG indicators defined by the EU comprises around 100 indicators. 31 indicators are used to monitor more than one goal. The indicators have been selected to take into account the EU context and perspective, availability, country coverage, data freshness and quality [5, 55].

Some authors have proposed approaches focusing on individual indicators. For example, [56] conducted a detailed assessment of the indicator 6.4.2 (i.e., Level of water stress). Firoiu et al. [57] used methods of dynamic analysis and prediction tools to assess progress on the SDGs achieved by Romania. Bidarbakhtnia [58] analysed three methods used by, respectively, OECD, the Sustainable Development Solutions Network (SDSN) and the United Nations Economic and Social Commission for Asia and the Pacific (UNESCAP). All of them measure the distance of each indicator from the 2030 target. The study conducted by UNESCAP also shows progress on each indicator since 2000 in proportion to total progress needed for the region to reach the 2030 target. Giupponi et al. [59] presented an approach for the spatial assessment of Water Use Efficiency (SDG indicator 6.4.1). Moyer and Hedden [60] used an integrated assessment model to evaluate progress toward target values on nine indicators related to six SDGs related to human development.

The use of individual indicators without an accurate and scientific follow up on their operationalization is difficult because of comparability issues, reporting requirements and decision-making processes [61–63]. Since there are studies indicating that relying exclusively on the global set of individual indicators leads to questionable results, some researchers argue that progress on SDGs should be additionally measured by means of composite indicators (e.g. [52, 64]. It is worth adding that in the case of many indicators, it is difficult to conduct analysis and evaluate the results for a large number of countries without taking into account some form of index aggregation even if a synthetic measure is hard to construct and causes some loss of information. Despite limitations, the literature describes some methods to assess progress on the SDGs, which involve composite indices and aggregated dashboards. Schmidt-Traub et al. [65] introduced the SDG Index, which synthesizes country-level data for all 17 SDGs taking into account the upper and lower bounds based on best and worst performing countries. The SDG Index can be used to estimate the distance that separates each country from achieving the SDGs. This approach is further developed in the article by [50], who propose a novel approach combining well-known methods to produce a comprehensive assessment of Australia's progress on all SDGs. An attempt to use a composite indicator called 'SDG achievement index' (SDG-AI) to measure the SDGs, covering six dimensions of sustainable development (Health, Education, Services, Employment, Equality and Environment) can be found in [66, 67]. The

SDG-AI can be used in two ways: to highlight differences between countries and to evaluate the contribution of different dimensions to the final result. Dhaoui [66] assessed progress on SDGs in MENA (Middle East and North Africa) countries. Rocchi et al. [67] modified the approach proposed by [66] to make it suitable for the EU context. A methodology for assessing SDGs on the aggregate level without losing information on single goals was proposed by [68]. Their study proposes several composite indices to assess the performance of EU member states. The two-stage approach involving Principal Component Analysis was applied to construct goal-based indices, pillar-based indices, and the overall SDG index. The indices were used to determine where a country currently stands on each of the indicators considered in the analysis, but they cannot be used to estimate the rate at which a country is moving towards achieving the SDGs.

Miola and Schiltz [14] compared three main approaches to measuring progress on SDGs at country level: the SDG Index, the OECD's distance measure, and progress measures based on Eurostat's report. They identified crucial weaknesses in these existing methods and their sensitivity to particular choices made along the way: depending on which indicators and approaches are applied, countries can receive substantially different relative evaluations. The authors identify the main methodological challenges that should be addressed when developing analytical tools to evaluate progress on SDGs.

Cavalli et al. [69] propose a model-based approach to evaluating the sustainability of the EU regional operational programme (ROP) in terms of SDGs, which is based on a synthetic sustainability index representing the part of ROP resources that contribute to the 2030 Agenda in relation to the total ROP resources. The usefulness of composite indicators has been demonstrated by [10], who compared the utility of the Multiple Reference Point Weak-Strong Composite Indicators (MRP-WSCI) and its partially compensatory version (MRP-PCI) for assessing the sustainability of EU countries according to the framework of the 2030 Agenda. The approach was used to produce composite indicators with different degrees of compensation, which constituted the basis for a country ranking.

Different variants of the methodology have already been used to build composite sustainability indicators in relation to SDG 7. Vavrek and Chovancová [13] assessed EU countries using a set of eight energy-related indicators. Indicator weights were determined using the coefficient of variation from the TOPSIS method. The authors assessed whether a country's performance resulted from a single indicator regarded as typical for the positively or negatively evaluated countries, or from a combination of indicators reflecting general energy issues. Chovancová and Vavrek [11] presented a continuation of their research, in which they identified the best and the worst performing EU countries taking into account a set of indicators.

Cheba and Bąk [12] proposed a synthetic measure based on the TOPSIS method to evaluate the relationship between SDG 7 and environmental production efficiency, which is a key component of the idea of green growth. They found considerable discrepancies between development paths of different EU countries despite their efforts to equalize the level of development in this area.

Dmytrów et al. [9] proposed an approach to assessing the EU's progress on SDG 7 at the national level using a synthetic measure obtained by applying the method of complex proportional assessment (COPRAS). They produced a ranking of countries in terms of their progress on SDG 7 by applying the Dynamic Time Warping method. Hierarchical clustering was then used to determine homogeneous groups of countries.

The rate of progress on SDG 7 achieved by the EU countries was also analysed by [49], who applied hierarchical cluster analysis to identify hidden associative structures. They also ranked EU countries in relation to the goals of the 2030 and 2050 Agenda. The ranking was used to identify clusters of countries sharing similar characteristics regarding their performance on

SDG 7. Cluster analysis was also used by [70, 71] to assess energy-related indicators of SDGs. In both studies, countries' performance was analysed and compared taking into account their own conditions and progress on SDG 7 and energy transformation processes taking place in EU member states.

While the importance of assessing progress towards achieving sustainable development goals is recognized, the number of articles proposing new methods, especially those representing a dynamic approach, is still rather limited. In the case of the SDGs, given the large number of indicators, it is reasonable to opt for composite indicators [10]. Composite indicators are associated with different degrees of compensation. One can distinguish fully compensatory, partially compensatory, or non-compensatory indicators (see section 1.2). The recommended approach is to assess sustainability in relation to a threshold or a target [10, 72]. According to [73], 'a given indicator does not say anything about sustainability, unless a reference value such as thresholds is given to it'. Non-compensatory methods can be used to identify weak points in the global assessment of an object and thus possible areas for improvement. For these reasons, we propose a novel non-compensatory approach based on principles of dynamic relative taxonomy applied to the procedure of constructing an aggregate measure. It can account for reference levels of each indicator and be used to rank countries accordingly showing the varying distance that separates individual EU countries from the targets set out in Agenda 2030. An additional advantage of this method is that the dynamic approach indicates not only relations between the objects in specific periods, but also changes in the phenomenon of interest that took place between objects over the entire reference period. Thus, it can be used to track cross-sectional and longitudinal changes.

## 3. Using dynamic relative taxonomy to construct a composite index

The classic approach to relative taxonomy was proposed by [74]. Lira [75] developed its positional version. Both approaches are static which means that relativization given by formula (5) is conducted separately for each single year of period analysed in the study. Static relative taxonomy has been applied, among others, by [76–78].

The following study involves the use of dynamic relative taxonomy, described in [21]. In the dynamic version, values of the $j$-th variable in formula (5) is relativised based on a matrix of data from all periods. This approach was extended by [22] to include robust measures of central tendency. Geometric mean was used in steps 6 and 7 of the dynamic relative taxonomy procedure (c.f. [16]):

1. Observations of $m$ variables for $n$ objects and $T + 1$ periods (2010–2021 and the year 2030, for which target values are set) are combined into one data matrix:

$$\left[y_{ijt}\right]_{n \cdot T^* \times m} = \begin{bmatrix} y_{111} & y_{121} & \cdots & y_{1m1} \\ \vdots & \vdots & \cdots & \vdots \\ y_{n11} & y_{n21} & \cdots & y_{nm1} \\ \cdots & \cdots & \cdots & \cdots \\ y_{11T} & y_{12T} & \cdots & y_{1mT} \\ \vdots & \vdots & \cdots & \vdots \\ y_{n1T} & y_{n2T} & \cdots & y_{nmT} \\ y_{11T^*} & y_{12T^*} & \cdots & y_{1mT^*} \\ \vdots & \vdots & \cdots & \vdots \\ y_{n1T^*} & y_{n2T^*} & \cdots & y_{nmT^*} \end{bmatrix}, \quad (1)$$

where: $i = 1, \ldots, n$—object number ($n = 28$: the EU as a whole and 27 EU countries),

$j = 1, \ldots, m$—variable number ($m = 7$: indicators for SDG 7 –see Table 2),
$t = 1, \ldots, T, T^*$—where $t = 1, \ldots, T$ represents years 2010–2021, and $t = T^*$ the year 2030, for which target values are set–see Table 2.

2. Stimulants and destimulants are identified in the set of variables (both terms were introduced by [79]). Instead of describing variables as stimulants and destimulants, [80] use the terms 'positive polarity' (increasing values of the index correspond to an improvement in the phenomenon of interest) and 'negative polarity' (increasing values of the index correspond to a deterioration in the phenomenon of interest). Hwang and Yoon [81, p. 130] use the concepts of 'benefit' (larger values of a variable are preferred) and 'cost' (larger values of a variable are less preferred).

3. Observations on each variable are replaced with target values if the following conditions are satisfied (data adjustment):

$$x_{ijt} = \begin{cases} y_{ijT^*} & \text{for} \quad y_{ijt} > y_{ijT^*} \\ y_{ijt} & \text{for} \quad y_{ijt} \leq y_{ijT^*} \end{cases}, \text{ for stimulants} \tag{2}$$

$$x_{ijt}^D = \begin{cases} y_{ijT^*} & \text{for} \quad y_{ijt} < y_{ijT^*} \\ y_{ijt} & \text{for} \quad y_{ijt} \geq y_{ijT^*} \end{cases}, \text{ for destimulants} \tag{3}$$

$y_{ijT^*}$—target values of SDG 7 indicators set for 2030.
For each variable, values higher (for stimulants) or lower (for destimulants) than the targets are replaced with the values of EU-level targets (target values of SDG 7 indicators set for 2030). This operation can be called one-sided Winsorization of the data (see e.g., [82]).

4. Destimulants $D$ are converted into stimulants using the ratio transformation:

$$x_{ijt} = \left( x_{ijt}^D \right)^{-1} \tag{4}$$

5. Values of each $j$-th variable are relativized according to the following $n \cdot T^* \times n \cdot T^*$ matrix:

$$\begin{bmatrix} 1 & \cdots & x_{njT^*}/x_{1j1} \\ \vdots & \vdots & \vdots \\ x_{1j1}/x_{nj1} & \cdots & x_{njT^*}/x_{nj1} \\ \cdots & \cdots & \cdots \\ x_{1j1}/x_{1jT} & \cdots & x_{njT^*}/x_{1jT} \\ \vdots & \vdots & \vdots \\ x_{1j1}/x_{njT} & \cdots & x_{njT^*}/x_{njT} \\ x_{1j1}/x_{1jT^*} & \cdots & x_{njT^*}/x_{1jT^*} \\ \vdots & \vdots & \vdots \\ x_{1j1}/x_{njT^*} & \cdots & 1 \end{bmatrix} \tag{5}$$

As a result of relativization, variable values are dimensionless. When the numerator is not greater than the denominator, the relativization formula produces values included in the interval $(0; 1]$, otherwise, values are included in the interval $(1; \infty)$.

6. The average similarity of a given relativized observation with respect to other relativized observations of the $j$-th variable for each column of matrix (5) is calculated using the

geometric mean:

$$\left[z_{ijt}\right] = \begin{bmatrix} \sqrt[n \cdot T^*]{\prod_{t=1}^{T^*} \prod_{i=1}^{n} \dfrac{x_{111}}{x_{i1t}}} & \cdots & \sqrt[n \cdot T^*]{\prod_{t=1}^{T^*} \prod_{i=1}^{n} \dfrac{x_{1m1}}{x_{imt}}} \\ \vdots & \vdots & \vdots \\ \sqrt[n \cdot T^*]{\prod_{t=1}^{T^*} \prod_{i=1}^{n} \dfrac{x_{n1T^*}}{x_{i1t}}} & \cdots & \sqrt[n \cdot T^*]{\prod_{t=1}^{T^*} \prod_{i=1}^{n} \dfrac{x_{nmT^*}}{x_{imt}}} \end{bmatrix} \tag{6}$$

The $[z_{ijt}]$ matrix is equivalent to a normalised matrix in multivariate statistical analysis.

7. Values of the composite indicator $SM_{it}$ are calculated according to the following formula:

$$SM_{it} = \sqrt[m]{\prod_{j=1}^{m} \frac{1}{z_{ijt}}} \tag{7}$$

Values of the composite indicator $SM_{it}$ given by (7) can be greater or smaller than 1. The smaller the value of $SM_{it}$ is, the better the position of object $i$ relative to other objects in a time interval from $t = 1$ to $t = T^*$. Unlike the static approach, the dynamic approach shows not only relations between the objects in specific periods, but also changes in the phenomenon of interest that took place between objects over the entire reference period.

The method of dynamic relative taxonomy is characterised by the following properties:

- it can only be used for variables measured on the ratio scale (their values are positive real numbers). In other words, it cannot be applied to interval-valued data. This is not a serious limitation since the majority of variables describing economic phenomena are measured on the ratio scale;

- composite indicators $SM_{it}$ do not have an upper bound, which does not disqualify them as such;

- the data matric can have missing values ($NA$), which are excluded in the calculation of the composite indicators $SM_{it}$;

- the disadvantage of the static approach when calculating the average similarity of a given relativized observation in relation to other relativized observations has been eliminated (more details can be found in [21].

## 4. Results in relation to EU-level targets for 2030

In the first step we analysed changes in $SM_{it}$ representing progress on SDG 7 in the EU countries. Table 3 shows values of $SM_{it}$ representing progress on SDG 7 achieved by EU countries in 2010–2021. The lower the value of $SM_{it}$, the better the position of object $i$ relative to other objects in each year and over the entire reference period. The dynamic approach reveals not only relationships between objects in different years but also changes that took place in the level of a given indicator over the entire reference period.

The last three columns in Table 3 show, respectively, the increment in the composite indicator between 2030 and 2010 ($\Delta = SM_{i2030} - SM_{i2010}$), the increment in the composite indicator between 2021and 2010 ($\Delta 1 = SM_{i2021} - SM_{i2010}$) and the distance of each EU country in relation to the target set for 2030 ($\Delta 2 = SM_{i2030} - SM_{i2021}$).

Countries furthest away from the target at the start of the reference period included Malta ($SM_{i2010} = 2.04$), Cyprus ($SM_{i2010} = 1.59$), Bulgaria ($SM_{i2010} = 1.52$), Belgium ($SM_{i2010} = 1.40$),

**Table 3. $SM_{it}$ values in relation to the EU-level target for 2030, representing progress on SDG 7 achieved from 2010 to 2021, and sorted by values observed in 2021.**

| No | Country | Values of the aggregate measure $SM_{it}$ | | | | | | | | | | | | | | |
|---|---|---|---|---|---|---|---|---|---|---|---|---|---|---|---|---|
| | | 2010 | 2011 | 2012 | 2013 | 2014 | 2015 | 2016 | 2017 | 2018 | 2019 | 2020 | 2021 | Δ | Δ1 | Δ2 |
| 1 | Sweden | 0.7967 | 0.7619 | 0.7448 | 0.7408 | 0.7243 | 0.7087 | 0.7608 | 0.7162 | 0.7304 | 0.7028 | 0.7209 | 0.6775 | −0.2481 | −0.1192 | −0.1289 |
| 2 | Denmark | 0.7661 | 0.7544 | 0.7443 | 0.7827 | 0.7271 | 0.7605 | 0.7289 | 0.7192 | 0.7260 | 0.7537 | 0.7726 | 0.7371 | −0.2175 | −0.0290 | −0.1885 |
| 3 | Estonia | 0.9878 | 0.9439 | 0.9754 | 0.9461 | 0.8596 | 0.8148 | 0.9100 | 0.9080 | 0.8695 | 0.8225 | 0.8146 | 0.7578 | −0.4392 | −0.2300 | −0.2092 |
| 4 | Austria | 0.9360 | 0.8791 | 0.8837 | 0.8673 | 0.8647 | 0.8437 | 0.8594 | 0.8535 | 0.8028 | 0.8221 | 0.7751 | 0.7869 | −0.3874 | −0.1491 | −0.2383 |
| 5 | Slovenia | 1.0198 | 1.0313 | 1.0381 | 0.9729 | 0.9474 | 0.9770 | 0.9637 | 0.9422 | 0.9069 | 0.8445 | 0.8184 | 0.7919 | −0.4712 | −0.2279 | −0.2433 |
| 6 | Finland | 0.9174 | 0.8922 | 0.8701 | 0.8540 | 0.8517 | 0.8310 | 0.8417 | 0.8471 | 0.8471 | 0.8313 | 0.8013 | 0.8163 | −0.3688 | −0.1011 | −0.2677 |
| 7 | Netherlands | 1.0763 | 0.9734 | 1.0043 | 1.0128 | 0.9486 | 1.0427 | 1.0117 | 0.9956 | 0.9767 | 0.9923 | 0.8809 | 0.8980 | −0.5277 | −0.1783 | −0.3494 |
| 8 | Latvia | 1.2351 | 1.2533 | 1.2270 | 1.1998 | 1.0983 | 1.0879 | 1.0340 | 1.0230 | 0.9965 | 0.9924 | 0.9253 | 0.9013 | −0.6865 | −0.3338 | −0.3527 |
| 9 | Germany | 1.1174 | 1.0691 | 1.0527 | 1.0897 | 1.0209 | 0.9949 | 0.9895 | 0.9658 | 0.9157 | 0.9048 | 0.9990 | 0.9069 | −0.5688 | −0.2105 | −0.3583 |
| 10 | France | 1.0700 | 1.0675 | 1.0520 | 1.0686 | 0.9876 | 0.9866 | 0.9721 | 0.9627 | 0.9449 | 0.9612 | 0.9077 | 0.9266 | −0.5214 | −0.1434 | −0.3780 |
| 11 | Czechia | 1.1068 | 1.1196 | 1.0851 | 1.0767 | 1.0416 | 1.0252 | 0.9946 | 0.9914 | 0.9601 | 0.9578 | 0.9014 | 0.9367 | −0.5582 | −0.1701 | −0.3881 |
| 12 | Portugal | 1.1676 | 1.1381 | 1.1201 | 1.1055 | 1.0768 | 1.0826 | 1.0516 | 1.0618 | 1.0457 | 1.0307 | 0.9469 | 0.9413 | −0.6190 | −0.2263 | −0.3927 |
| 13 | Croatia | 1.0719 | 1.0864 | 1.0592 | 1.0295 | 0.9812 | 1.0305 | 1.0145 | 1.0120 | 0.9967 | 0.9781 | 0.9306 | 0.9524 | −0.5233 | −0.1195 | −0.4038 |
| 14 | Ireland | 1.2918 | 1.2206 | 1.2210 | 1.2445 | 1.1764 | 1.1966 | 1.1020 | 1.0234 | 1.0360 | 1.0357 | 0.9261 | 0.9659 | −0.7432 | −0.3259 | −0.4173 |
| 15 | EU | 1.1571 | 1.1282 | 1.1257 | 1.1084 | 1.0564 | 1.0601 | 1.0511 | 1.0386 | 1.0198 | 0.9965 | 0.9584 | 0.9693 | −0.6085 | −0.1878 | −0.4207 |
| 16 | Romania | 1.0715 | 1.0371 | 1.0214 | 0.9742 | 0.9370 | 0.9457 | 0.9451 | 0.9387 | 0.9160 | 0.9305 | 0.9294 | 0.9849 | −0.5229 | −0.0866 | −0.4363 |
| 17 | Italy | 1.1949 | 1.2326 | 1.2278 | 1.1715 | 1.1069 | 1.1369 | 1.1149 | 1.1031 | 1.0861 | 1.0417 | 0.9467 | 0.9870 | −0.6463 | −0.2079 | −0.4384 |
| 18 | Luxembourg | 1.2730 | 1.2470 | 1.2161 | 1.1779 | 1.1119 | 1.1125 | 1.0939 | 1.0816 | 1.0393 | 1.0835 | 1.0308 | 0.9885 | −0.7244 | −0.2845 | −0.4399 |
| 19 | Spain | 1.0418 | 1.0092 | 1.0377 | 0.9783 | 1.0139 | 1.0265 | 0.9950 | 0.9795 | 0.9993 | 0.9559 | 0.9294 | 0.9926 | −0.4932 | −0.0492 | −0.4440 |
| 20 | Greece | 1.2219 | 1.2669 | 1.2691 | 1.1608 | 1.1836 | 1.2351 | 1.2128 | 1.1738 | 1.1113 | 1.0739 | 1.0472 | 1.0315 | −0.6733 | −0.1904 | −0.4829 |
| 21 | Poland | 1.3553 | 1.3026 | 1.2612 | 1.1900 | 1.1316 | 1.1047 | 1.1305 | 1.1614 | 1.1405 | 1.0790 | 1.0124 | 1.0342 | −0.8067 | −0.3211 | −0.4856 |
| 22 | Slovakia | 1.1522 | 1.0905 | 1.0949 | 1.1059 | 1.0590 | 1.0473 | 1.0369 | 1.0554 | 1.0486 | 1.1207 | 1.0223 | 1.0581 | −0.6036 | −0.0941 | −0.5095 |
| 23 | Belgium | 1.4024 | 1.3483 | 1.3030 | 1.2975 | 1.2070 | 1.2320 | 1.2069 | 1.2249 | 1.2055 | 1.1344 | 1.0649 | 1.0735 | −0.8538 | −0.3289 | −0.5249 |
| 24 | Hungary | 1.2931 | 1.2673 | 1.2534 | 1.2232 | 1.1959 | 1.1882 | 1.1974 | 1.1927 | 1.1541 | 1.1523 | 1.0677 | 1.1215 | −0.7445 | −0.1716 | −0.5729 |
| 25 | Cyprus | 1.5876 | 1.5589 | 1.5294 | 1.4055 | 1.3668 | 1.4245 | 1.3962 | 1.3885 | 1.3077 | 1.3044 | 1.1956 | 1.1720 | −1.0390 | −0.4156 | −0.6234 |
| 26 | Malta | 2.0382 | 1.9016 | 1.8416 | 1.7563 | 1.6676 | 1.5178 | 1.3306 | 1.3475 | 1.3660 | 1.3791 | 1.2420 | 1.2217 | −1.4896 | −0.8165 | −0.6731 |
| 27 | Lithuania | 1.3615 | 1.4123 | 1.3874 | 1.3060 | 1.2677 | 1.2800 | 1.2989 | 1.3048 | 1.3328 | 1.3014 | 1.2494 | 1.2733 | −0.8129 | −0.0882 | −0.7247 |
| 28 | Bulgaria | 1.5183 | 1.4553 | 1.4140 | 1.3405 | 1.3329 | 1.3513 | 1.3470 | 1.3560 | 1.2995 | 1.2671 | 1.2306 | 1.2827 | −0.9697 | −0.2356 | −0.7341 |
| | mean27 | 1.1879 | 1.1600 | 1.1457 | 1.1140 | 1.0699 | 1.0735 | 1.0571 | 1.0493 | 1.0282 | 1.0168 | 0.9663 | 0.9710 | | | |
| | sd27 | 0.2548 | 0.2436 | 0.2307 | 0.2069 | 0.1981 | 0.1909 | 0.1677 | 0.1741 | 0.1695 | 0.1671 | 0.1421 | 0.1535 | | | |
| | range27 | 1.2721 | 1.1472 | 1.0973 | 1.0155 | 0.9433 | 0.8091 | 0.6673 | 0.6723 | 0.6400 | 0.6763 | 0.5285 | 0.6052 | | | |

$SM_{i2030}$ = 0.5486 –target value of the aggregate measure (EU 2030) calculated on the basis of the indicators used to measure progress on SDG; Δ = $SM_{i2030}$ − $SM_{i2010}$; Δ1 = $SM_{i2021}$ − $SM_{i2010}$; Δ2 = $SM_{i2030}$ − $SM_{i2021}$; Δ = Δ1 + Δ2; mean27, sd27, and range27 –arithmetic mean, standard deviation and range for the 27 EU countries.
Source: Authors' calculations using R program [83].

Lithuania ($SM_{i2010}$ = 1.36), Poland ($SM_{i2010}$ = 1.36). Countries that were closest to the targets in 2010 included three Scandinavian–Denmark ($SM_{i2010}$ = 0.77), Sweden ($SM_{i2010}$ = 0.80), Finland ($SM_{i2010}$ = 0.92) and Austria ($SM_{i2010}$ = 0.94). Between 2010 and 2021 values of $SM_{it}$ fell on average by nearly 2% each year, which means that the distance of all countries from the targets kept decreasing. The biggest average annual change (decline) during the reference period was observed for Malta (4.5%), Latvia (2.8%), Cyprus (2.7%) and Ireland (2.6%). The smallest average annual change (decline) throughout the reference period was observed for Denmark (0.4%), Finland (1.1%) and Sweden (1.5%).

The distance of the European Union as a whole in 2021 in relation to the target set for 2030 is Δ1 = −0.4207. Fourteen EU countries were closer to the target in 2021 (see Fig 1). Because of

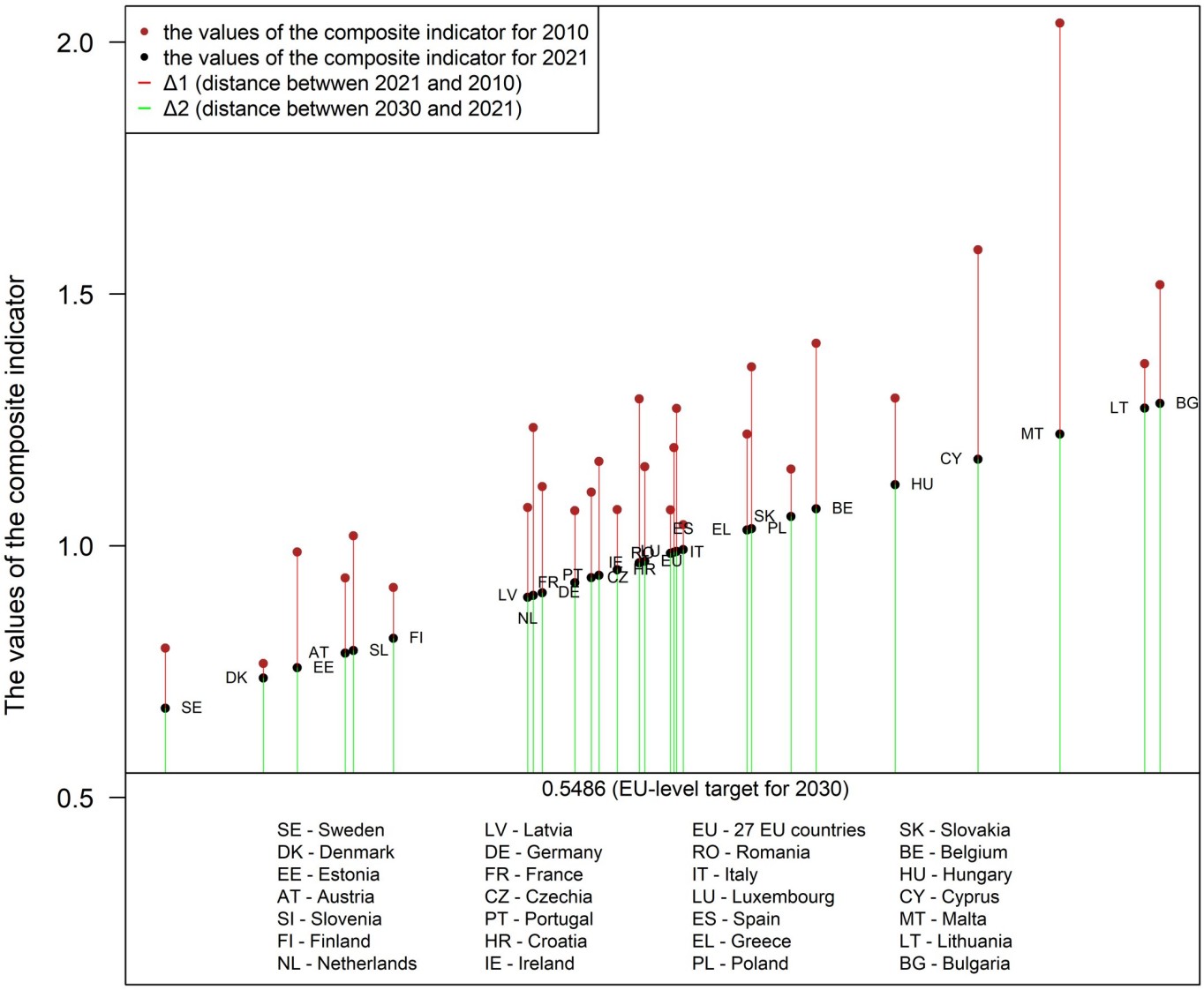

**Fig 1. A graphical representation of $SM_{it}$ values in relation to the EU-level target for 2030, representing progress on SDG 7 achieved from 2010 to 2021, and sorted by values observed in 2021.** Source: Chart created using R program [83].

changes in the values of the seven indicators that took place over 11 years, the group of countries that came closest to the 2030 target in 2021, apart from the three Scandinavian countries, includes Estonia, Austria, and Slovenia. Despite big increases in the values of the composite indicator, the largest distance in relation to the SDG 7 target in 2021 can be observed for Bulgaria, Lithuania, Malta, and Cyprus.

Fig 1 provides a graphical representation of $SM_{it}$ values in relation to the EU-level target for 2030, representing progress on SDG 7 achieved from 2010 to 2021, and sorted by values observed in 2021. The horizontal line represents the EU-level SDG 7 target value of the composite indicator $SM_{i2030} = 0.5486$.

The biggest progress (improvement in the value of $SM_{it}$ between 2010 and 2021) on SDG 7 in the period 2010–2021 was made by Malta ($\Delta1 = -0.8165$), followed by Cyprus ($\Delta1 = -0.4156$), Latvia ($\Delta1 = -0.3338$), Belgium ($\Delta1 = -0.3289$), Ireland ($\Delta1 = -0.3259$) and

Poland ($\Delta 1 = -0.3211$). The European Union as a whole is making progress towards meeting the SDG 7 goal. The improvement in the value of the aggregate measurer $SM_{it}$ in the period 2010–2021 was $\Delta 1 = -0.1878$. In the Eurostat report [84], progress towards the SDG 7 goal was described as moderately favorable.

Compared to 2010, the biggest climb in the ranking for 2021 can be observed for Latvia (by 11 places, from 19th to 8th), Ireland (by 7 places, from 21st to 14th), Netherlands (by 4 places, from 11th to 7th), Germany (by 4 places, from 13th to 9th), Portugal (by 4 places, from 16th to 12th). Because of small improvement in the value of the composite indicator for Spain ($\Delta 1 = -0.0492$), Romania ($\Delta 1 = -0.0866$) and Slovakia ($\Delta 1 = -0.0941$), they fell in the ranking, respectively, by 12 places (from 7th to 19th), by 7 places (from 9th to 16th) and by 8 places (from 14th to 22nd).

It is worth noting that in 2021 as many as 12 countries represent a similar, average level of the composite indicator, ranging from 0.898 (Netherlands) to 0.993 (Spain). The degree of variation between the smallest and the biggest values is considerably greater.

Fig 2 includes a line graph showing progress on SDG 7 achieved by the 28 objects (the EU and the 27 EU countries) between 2010 and 2021.

The reference period includes the time of the COVID-19 pandemic, which evidently affected deterioration on SDG 7 achieved by Sweden, Denmark, and Germany in 2020 in relation to the performance observed in 2019. A similar deterioration in performance can be observed in 2021 in relation to 2020 in the case of 16 countries (Austria, Finland, Netherlands, France, Czech Republic, Croatia, Ireland, Romania, Italy, Spain, Poland, Slovakia, Belgium, Hungary, Lithuania, Bulgaria). No negative effects during the pandemic can be observed for the remaining 8 countries.

The horizontal line at the bottom of the chart represents the EU-level target for 2030: $SM_{i2030} = 0.5486$. A systematic decrease both the average value of the aggregate measure and its diversification can be observed in the entire period under study (see Table 3, Fig 2). First of all, this indicates systematic progress on SDG 7 achieved by the EU countries. Secondly, it shows that differences between EU countries keep getting smaller, as evidenced by the range and the standard deviations of the composite indicator (see Table 3). The range of the composite indicator decreased from $R_{2010}^{SM} = 1.2721$ in 2010 to $R_{2021}^{SM} = 0.6052$ in 2021.

## 5. Conclusions and policy implications

Since September 2015, 193 UN countries have been working to implement the resolution "Transforming our world: the 2030 Agenda for Sustainable Development" containing 17 Sustainable Development Goals (SDGs). To ensure sustainable performance in the context of the challenges posed by the 2030 Agenda, each EU country needs to properly assess its progress towards achieving SDGs. Given the importance of this need, we set out to analyse the performance of the EU countries in relation to the core targets covered by SDG 7, measuring progress by using seven indicators. For three of these indicators the European Commission has set target values for 2030. For the other four, the target values were based on the "best performance" among the EU countries in 2015.

Progress on SDGs is typically measured by means of composite indicators. Three methodological approaches can be distinguished depending on the degree of compensation: fully compensatory, partially compensatory, and non-compensatory. The third approach is recommended since in addition to accounting for target values of the indicators, it involves data adjustment and can be used to identify weak points in the country's global assessment, and, consequently, indicate possible areas for improvement (see section 1.2). Because the main goal of the study was to assess progress on SDG 7 achieved by individual EU countries and to

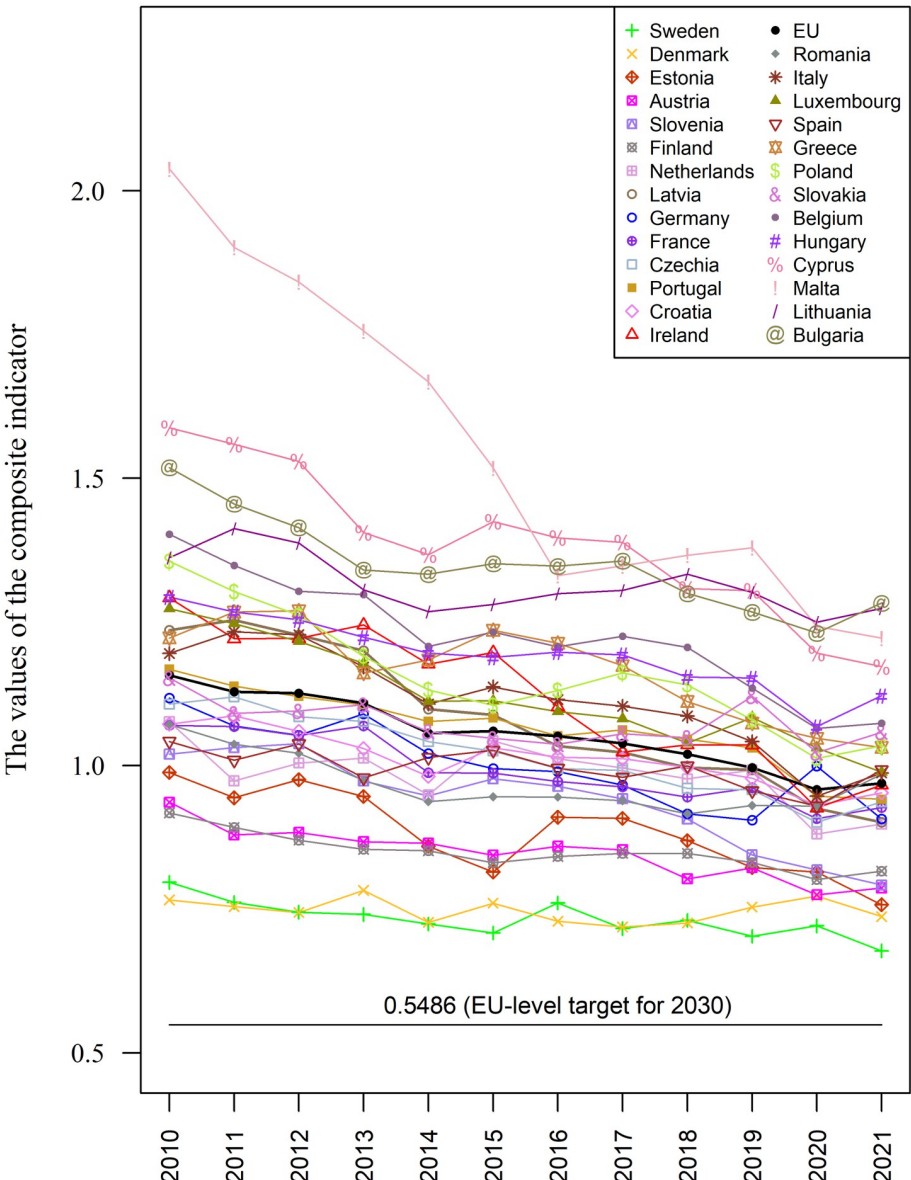

**Fig 2. Changes in progress on SDG 7 achieved by the EU countries between 2010 and 2021 in relation to the EU-level target for 2030.** Source: Chart created using R program [83].

determine their distance in relation to the target set for 2030, we decided to use a multidimensional indicator proposed and constructed by applying dynamic relative taxonomy (section 3). The method used in the analysis represents a non-compensatory approach.

During the reference period (2010–2021), all EU countries made systematic progress towards achieving SDG 7, although to a different degree and at different rate. Looking at how many countries managed to achieve the EU targets for 2030, the best result can be observed for the x7 indicator (Share of population unable to keep home adequately warm), which was achieved by Sweden, Finland, Slovenia, Austria; followed by the x5 indicator (Share of renewable energy in gross final energy consumption)–achieved by Sweden, Finland, Latvia; the x4 indicator (Energy productivity)–achieved by Denmark, Ireland, Luxembourg; for the x3

indicator (Final energy consumption in households per capita)–achieved by Spain, Malta, Portugal; for the x6 indicator (Energy import dependency)–achieved by Sweden, Estonia. Greece was the only country to achieve the 2030 target level for the remaining two indicators, namely x1 (Primary energy consumption) and x2 (Final energy consumption).

In 2010, the distance from the 2030 target was the biggest for Malta, Cyprus, Bulgaria, Belgium, Lithuania, Poland, and the smallest for Denmark, Sweden, Finland, and Austria. Because of changes in the values of the seven indicators that took place over 11 years, the group of countries that came closest to the 2030 target in 2021, apart from the three Scandinavian countries, includes Estonia, Austria, and Slovenia. Despite big increases in the values of the composite indicator, Bulgaria, Lithuania, Malta, and Cyprus remained furthest away from the SDG 7 target in 2021.

The trend regarding the implementation of the 2030 Agenda has been influenced by the COVID-19 pandemic. Many EU countries experienced a slowdown in the progress towards SDGs. In the case of SDG 7, the distance from the 2030 target in 2020 compared to 2019 had increased for Sweden, Denmark, and Germany. A similar deterioration in performance can be observed from 2020 to 2021 in the case of 16 countries (Austria, Finland, Netherlands, France, Czechia, Croatia, Ireland, Romania, Italy, Spain, Poland, Slovakia, Belgium, Hungary, Lithuania, Bulgaria). It is worth emphasizing that during the pandemic, some improvement could be observed for certain indicators. Counter-pandemic measures helped to enhance energy efficiency, which is one of the key pillars in achieving SDG 7. The restrictions on public life and lower economic activity reduced energy consumption from 2019 to 2020 by more than 8%. The reduction in final energy consumption also resulted in greater energy supply and increased the share of renewables in gross final energy consumption. The economic recovery of 2021 and the return to pre-pandemic mobility patterns increased the demand for energy again. However, consumption remained below pre-pandemic levels as the effects of the pandemic continued to shape energy and economic activities. The most obvious negative consequence of the pandemic was the increase in energy consumption by EU households. Given these short-term trends, it is clear that in order to ensure the EU achieves its goals by 2030, changes in all indicators should be continuously monitored and assessed. The main motivation of our study was therefore to propose a methodological approach that could provide information to make such monitoring possible.

The novelty of the study consists in applying a non-compensatory approach to show the varying distance that separates individual EU countries from the targets set out in Agenda 2030. This was possible thanks to the use of the aggregate approach in the methodology of dynamic relative taxonomy (see section 3) taking into account the target values of the indicators for 2030 with data adjustment (one-sided Winsorization in step 3 –see section 1.2) and the use of the geometric mean in step 6 and 7 of the procedure of the aggregate measure constructing. It is worth noting that the dynamic approach indicates not only relations between the objects in specific periods, but also changes in the phenomenon of interest that took place between objects over the entire reference period. In other words, it can be used to tracking changes from a cross-sectional and longitudinal perspective.

The conducted study has also its limitations. It was not possible to use all SDG 7 indicators included in Agenda 2030 because required statistical data were not directly comparable (the first two indicators in Table 2), as explained in section 1.2. The key disadvantage of aggregate methods, i.e., their compensatory nature (the fact that positive and negative deviations from the target values of individual indicators can accumulate), was considerably reduced by utilising the geometric mean and by applying one-sided Winsorization of the data, according to formulas (2) and (3).

The results of the study contribute to research on energy security, which is currently an essential element of the economic security system. The proposed approach involving dynamic relative taxonomy can be used as a tool supporting efforts to monitor progress on SDG 7 as part of the national energy policy. It is worth emphasizing that progress achieved by particular countries can be more relevant than their final performance outcomes. In order to properly assess progress regarding the goals of Agenda 2030, in addition to calculating indices and creating country rankings for selected years, it is necessary to analyse changes over time using the dynamic approach.

By design, all 17 SDGs are an integrated set of global priorities and objectives but SDG 7, as well as SDG 2 (Zero hunger), SDG 3 (Good health and well-being), SDG 14 (Life below water), are classified as being the most synergistic with other SDGSs. Therefore, the results presented in this article can be treated as a starting point for policy makers and other stakeholders interested in identifying the main directions of change, priorities, and strategies in national policies of sustainable development.

As demonstrated in the study, the proposed methodology can be used not only to assess progress on SDG 7 but it also provides a relevant contribution to research regarding methods of measuring national progress on the other SDGs included in Agenda 2030.

## Author Contributions

**Conceptualization:** Marek Walesiak, Grażyna Dehnel.

**Data curation:** Grażyna Dehnel.

**Formal analysis:** Marek Walesiak, Grażyna Dehnel.

**Investigation:** Grażyna Dehnel.

**Methodology:** Marek Walesiak, Grażyna Dehnel.

**Software:** Marek Walesiak.

**Validation:** Marek Walesiak, Grażyna Dehnel.

**Visualization:** Marek Walesiak.

**Writing – original draft:** Marek Walesiak, Grażyna Dehnel.

**Writing – review & editing:** Marek Walesiak, Grażyna Dehnel.

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
