## [Decision Letter · Decision Letter 0]

15 Sep 2023

PONE-D-23-21807Progress on SDG 7 achieved by EU countries in relation to the target year 2030:

a multidimensional indicator analysis using dynamic relative taxonomyPLOS ONE

Dear Dr. Dehnel,

Thank you for submitting your manuscript to PLOS ONE. After careful consideration, we feel that it has merit but does not fully meet PLOS ONE’s publication criteria as it currently stands. Therefore, we invite you to submit a revised version of the manuscript that addresses the points raised during the review process.

We look forward to receiving your revised manuscript.

Kind regards,

André Ramalho, PhD

Academic Editor

PLOS ONE

Journal Requirements:

2. We note that Figure 2 in your submission contain map/satellite images which may be copyrighted. All PLOS content is published under the Creative Commons Attribution License (CC BY 4.0), which means that the manuscript, images, and Supporting Information files will be freely available online, and any third party is permitted to access, download, copy, distribute, and use these materials in any way, even commercially, with proper attribution. For these reasons, we cannot publish previously copyrighted maps or satellite images created using proprietary data, such as Google software (Google Maps, Street View, and Earth). For more information, see our copyright guidelines: http://journals.plos.org/plosone/s/licenses-and-copyright.

Reviewers' comments:

Reviewer's Responses to Questions

**Comments to the Author**

1. Is the manuscript technically sound, and do the data support the conclusions?

Reviewer #1: Yes

2. Has the statistical analysis been performed appropriately and rigorously? 

Reviewer #1: Yes

3. Have the authors made all data underlying the findings in their manuscript fully available?

Reviewer #1: Yes

4. Is the manuscript presented in an intelligible fashion and written in standard English?

Reviewer #1: Yes

5. Review Comments to the Author

Reviewer #1: Dear Authors,

I write to you with a sense of great satisfaction after having thoroughly reviewed your manuscript. I find your study notably valuable, both for its novel application of a non-compensatory approach and its focus on the varying distance individual EU countries have from the targets set out in the Agenda 2030. The dynamic nature of your methodology, combined with the sophisticated use of data adjustment and geometric means, lends a robustness to the work that cannot be understated.

Your research astutely recognises and explores the inherent synergy between the various Sustainable Development Goals (SDGs). This broad, interconnected perspective provides a comprehensive understanding that is highly beneficial to this field of study. I am particularly impressed by the way the study's findings extend beyond just assessing progress on SDG 7, but also illuminate potential pathways to address other Sustainable Development Goals. In light of your work, I would strongly encourage future research to further investigate these correlations between SDGs, as I believe it will yield significant insights in the pursuit of sustainable development.

I am confident that the insight you have provided will act as an effective foundation for policy makers and stakeholders in defining the main directions of change, priorities, and strategies in national sustainable development policies. Your work not only elucidates relations between the objects in specific periods but also uncovers changes in the phenomena of interest across the reference period, providing valuable cross-sectional and longitudinal perspectives.

Given the robustness of your study and its significant contribution to the field, I recommend your manuscript for publication. It serves as an excellent addition to the academic discourse surrounding Agenda 2030 and sustainable development at large.

Best Regards.

6. PLOS authors have the option to publish the peer review history of their article (what does this mean?). If published, this will include your full peer review and any attached files.

Reviewer #1: No

---

## [Author Response · Author response to Decision Letter 0]

14 Nov 2023

Answers to the comments of the Reviewer 1 on the paper:

“Progress on SDG 7 achieved by EU countries in relation to the target year 2030:

a multidimensional indicator analysis using dynamic relative taxonomy”

We would like to thank the Reviewer for the time and efforts necessary to review this paper and for favourable review of the article. We are glad that you found our research valuable and well orga-nized. Your positive assessment motivates and strengthens our commitment to further research in this area.

---

## [Decision Letter · Decision Letter 1]

23 Nov 2023

PONE-D-23-21807R1Progress on SDG 7 achieved by EU countries in relation to the target year 2030: a multidimensional indicator analysis using dynamic relative taxonomyPLOS ONE

Dear Dr. Dehnel,

Thank you for submitting your manuscript to PLOS ONE. After careful consideration, we feel that it has merit but does not fully meet PLOS ONE’s publication criteria as it currently stands. Therefore, we invite you to submit a revised version of the manuscript that addresses the points raised during the review process.

Please include the following items when submitting your revised manuscript:A rebuttal letter that responds to each point raised by the academic editor and reviewer(s). You should upload this letter as a separate file labeled 'Response to Reviewers'.A marked-up copy of your manuscript that highlights changes made to the original version. You should upload this as a separate file labeled 'Revised Manuscript with Track Changes'.An unmarked version of your revised paper without tracked changes. You should upload this as a separate file labeled 'Manuscript'.If applicable, we recommend that you deposit your laboratory protocols in protocols.io to enhance the reproducibility of your results. Protocols.io assigns your protocol its own identifier (DOI) so that it can be cited independently in the future. For instructions see: https://journals.plos.org/plosone/s/submission-guidelines#loc-laboratory-protocols. Additionally, PLOS ONE offers an option for publishing peer-reviewed Lab Protocol articles, which describe protocols hosted on protocols.io. Read more information on sharing protocols at https://plos.org/protocols?utm_medium=editorial-email&utm_source=authorletters&utm_campaign=protocols.

We look forward to receiving your revised manuscript.

Kind regards,

André Ramalho, PhD

Academic Editor

PLOS ONE

Journal Requirements:

Reviewers' comments:

Reviewer's Responses to Questions

**Comments to the Author**

1. If the authors have adequately addressed your comments raised in a previous round of review and you feel that this manuscript is now acceptable for publication, you may indicate that here to bypass the “Comments to the Author” section, enter your conflict of interest statement in the “Confidential to Editor” section, and submit your "Accept" recommendation.

Reviewer #2: (No Response)

Reviewer #3: (No Response)

2. Is the manuscript technically sound, and do the data support the conclusions?

Reviewer #2: Yes

Reviewer #3: Yes

3. Has the statistical analysis been performed appropriately and rigorously? 

Reviewer #2: Yes

Reviewer #3: Yes

4. Have the authors made all data underlying the findings in their manuscript fully available?

Reviewer #2: Yes

Reviewer #3: Yes

5. Is the manuscript presented in an intelligible fashion and written in standard English?

Reviewer #2: Yes

Reviewer #3: Yes

6. Review Comments to the Author

Reviewer #2: Authors did not address the fact that the expression “in which they identified the best and the worst performing countries” is repeated in the same sentence (lines 333-336) – “Chovancová and Vavrek [11] presented a continuation of their research, in which they identified the best and the worst performing countries in which they identified the best and the worst performing EU countries taking into account a set of indicators.”. However, I consider it still allows full comprehension of the conveyed message.

Reviewer #3: Review report on 'Progress towards SDG 7 achieved by EU countries towards the 2030 target year: a multivariate analysis of indicators using a dynamic relational taxonomy'

The manuscript under review is well written and provides a comprehensive assessment of EU countries' progress towards SDG 7. However, there are several areas that could be improved to enhance the quality and impact of the document. We make the following comments for the authors' consideration:

1. The introduction contains a lot of general information about SDG 7 and its indicators. To improve the introduction, the authors should focus on clearly stating the objective of the study. It would have been beneficial to add a few sentences that briefly introduce the research problem, objectives, novelty and significance of the study.

2. The literature review section is well written and provides a comprehensive overview of previous research. However, it is necessary to highlight the research gap that this study addresses. The authors should highlight why their study is important in relation to the existing literature. What specific gaps or limitations of previous studies does this research seek to overcome?

3. It is appropriate to include a composite indicator analysis for the EU as a whole. This analysis should provide insight into how the European Union as a whole is progressing towards SDG 7.

4. The authors should compare and discuss the results of their study with the results of other similar studies in the field of research. This will help readers to understand how the findings of this paper match or differ from existing research. Also, a discussion of the differences between countries will offer valuable insights into regional development.

5. In the final section, the authors should highlight the policy implications of their study findings. For example, they could highlight which indicators were most affected by events such as the pandemic and provide recommendations for policy responses or preventive measures. In addition, it is essential to discuss how development policies can be shaped to avoid a return to pre-crisis patterns. This will make the document more practical and valuable for policy makers.

7. PLOS authors have the option to publish the peer review history of their article (what does this mean?). If published, this will include your full peer review and any attached files.

Reviewer #2: **Yes: **Bruno Filipe Coelho da Costa

Reviewer #3: No

---

## [Author Response · Author response to Decision Letter 1]

5 Jan 2024

I have attached two files with precise responses to reviewers #2 and #3 in section "Attach Files"

Reviewer #2:

We would like to thank the Reviewer for valuable comment, which have been taken into account in the revised version of the article.

Please find below the answer to comment

Comment.

Authors did not address the fact that the expression “in which they identified the best and the worst performing countries” is repeated in the same sentence (lines 333-336) – “Chovancová and Vavrek [11] presented a continuation of their research, in which they identified the best and the worst per-forming countries in which they identified the best and the worst performing EU countries taking into account a set of indicators.”. However, I consider it still allows full comprehension of the conveyed message.

Answer. 

The remark has been taken into account in the corrected version of the article. The sentence has been reformulated into:

Chovancová and Vavrek [11] presented a continuation of their research, in which they identified the best and the worst performing EU countries in which they identified the best and the worst per-forming EU countries taking into account a set of indicators. 

Reviewer #3:

We would like to thank the Reviewer for valuable comments, which have been taken into account in the revised version of the article.

Please find below the answers to each comment

Comment 1.

The introduction contains a lot of general information about SDG 7 and its indicators. To improve the introduction, the authors should focus on clearly stating the objective of the study. It would have been beneficial to add a few sentences that briefly introduce the research problem, objectives, novelty and significance of the study.

Answer. 

The remark has been taken into account in the corrected version of the article. Section 1.2 introduces the following supplemented corrected text:

The purpose of the following study is to assess progress made by individual EU countries in 2010-2021 towards achieving SDG 7 and to determine their distance in relation to targets set for 2030. To overcome the problems with the first and second approach, we propose an innovative third approach, which accounts for the target values of the indicators and uses adjusted data. This approach has not been used in this type of research so far and the use of the geometric mean in the dynamic relative taxonomy method has made it possible to reduce the impact of the negative compensation effect on the values of aggregate measures. The results of the study have important implications for individual EU countries. In addition to showing progress towards the SDG 7 target, they also represent the distance that separates each country from achieving the the 2030 target. The proposed methodology can also be used to assess progress made by EU countries in the implementation of other SDGs of the 2030 Agenda.

Comment 2.

The literature review section is well written and provides a comprehensive overview of previous research. However, it is necessary to highlight the research gap that this study addresses. The authors should highlight why their study is important in relation to the existing literature. What specific gaps or limitations of previous studies does this research seek to overcome?

Answer. 

The remark has been taken into account in the corrected version of the article. Gaps and limitations of previous studies on the assessment of SDG7 are presented in Section 1.2. This is reflected in the revised version of the text with regard to note 1. In addition, the last paragraph of Section 2 has been replaced with the following text:

While the importance of assessing progress towards achieving sustainable development goals is recognized, the number of articles proposing new methods, especially those representing a dynamic approach, is still rather limited. In the case of the SDGs, given the large number of indicators, it is reasonable to opt for composite indicators [10]. Composite indicators are associated with different degrees of compensation. One can distinguish fully compensatory, partially compensatory, or non-compensatory indicators (see section 1.2). The recommended approach is to assess sustainability in relation to a threshold or a target [10, 72]. According to [73], ‘a given indicator does not say anything about sustainability, unless a reference value such as thresholds is given to it’. Non-compensatory methods can be used to identify weak points in the global assessment of an object and thus possible areas for improvement. For these reasons, we propose a novel non-compensatory approach based on principles of dynamic relative taxonomy applied to the procedure of constructing an aggregate measure. It can account for reference levels of each indicator and be used to rank countries accordingly showing the varying distance that separates individual EU countries from the targets set out in Agenda 2030. An additional advantage of this method is that the dynamic approach indicates not only relations between the objects in specific periods, but also changes in the phenomenon of interest that took place between objects over the entire reference period. Thus, it can be used to track cross-sectional and longitudinal changes.

Comment 3.

It is appropriate to include a composite indicator analysis for the EU as a whole. This analysis should provide insight into how the European Union as a whole is progressing towards SDG 7.

Answer. 

The remark has been taken into account in the corrected version of the article. 

In section 4 the following text has been added:

The European Union as a whole is making progress towards meeting the SDG 7 goal. The improvement in the value of the aggregate measurer 〖SM〗_it in the period 2010-2021 was Δ1=-0.1878. In the Eurostat report [84], progress towards the SDG 7 goal was described as moderately favorable.

In section 4 the following text, highlighted in red, has been added:

The distance of the European Union as a whole in 2021 in relation to the target set for 2030 is Δ1=–0.4207. Fourteen EU countries were closer to the target in 2021 (see Figure 1).

The following item has been added to the references:

84. Eurostat. Sustainable Development in the European Union: monitoring report on progress towards the SDGs in an EU context – 2023 edition. Luxembourg: Eurostat. 2023. https://ec.europa.eu/eurostat/web/products-flagship-publications/w/ks-04-23-184 (accessed 12 December 2023).

Comment 4.

The authors should compare and discuss the results of their study with the results of other similar studies in the field of research. This will help readers to understand how the findings of this paper match or differ from existing research. Also, a discussion of the differences between countries will offer valuable insights into regional development.

Answer. 

There a number of reasons why it is not possible to directly compare our results with those obtained in other studies (mentioned in the literature review in section 1.2 and section 2):

– different reference periods and sets of objects used in those studies,

– the inclusion of a larger or smaller number of variables than those used to measure progress on SDG 7 achieved by EU countries,

– different approaches used. Section 1.2 describes three approaches. Since the first approach does not account for the target values these indicators should achieve by 2030, it cannot be used to determine the distance of individual EU countries in relation to targets set for 2030. The second approach takes into account the target values of the indicators without data adjustment. The articles cited in section 1.2 present the first and second approaches. The disadvantages of the first approach include the fact that the target values of the indicators for 2030 are not taken into account (it is not possible to determine the distance of individual EU countries in relation to the target values); the data are not corrected – the values of aggregate measures are distorted (owing to the compensation effect). The use of non-corrected data is also the main disadvantage of the second approach. To overcome these problems, we propose an innovative third approach, which accounts for the target values of the indicators and uses adjusted data. The data are corrected as follows: when a given country exceeds the EU-level targets for 2030, these higher national values (in the case of stimulants) or lower national values (in the case of destimulants) are not included in the analysis but are replaced by target values set for the entire European Union. In this way, the drawbacks of the first and second approaches are eliminated.

Comment 5.

In the final section, the authors should highlight the policy implications of their study findings. For example, they could highlight which indicators were most affected by events such as the pandemic and provide recommendations for policy responses or preventive measures. In addition, it is essential to discuss how development policies can be shaped to avoid a return to pre-crisis patterns. This will make the document more practical and valuable for policy makers.

Answer. 

The remark has been taken into account in the corrected version of the article. The fifth paragraph in Section 5 has been replaced by the following text:

The trend regarding the implementation of the 2030 Agenda has been influenced by the COVID-19 pandemic. Many EU countries experienced a slowdown in the progress towards SDGs. In the case of SDG 7, the distance from the 2030 target in 2020 compared to 2019 had increased for Sweden, Denmark, and Germany. A similar deterioration in performance could be observed in 2021 in relation to 2020 from 2020 to 2021 in the case of 16 countries (Austria, Finland, Netherlands, France, Czechia, Croatia, Ireland, Romania, Italy, Spain, Poland, Slovakia, Belgium, Hungary, Lithuania, Bulgaria). It is worth emphasizing that during the pandemic, some improvement could be observed for certain indicators. Counter-pandemic measures helped to enhance energy efficiency, which is one of the key pillars in achieving SDG 7. The restrictions on public life and lower economic activity reduced energy consumption from 2019 to 2020 by more than 8%. The reduction in final energy consumption also resulted in greater energy supply and increased the share of renewables in gross final energy consumption. The economic recovery of 2021 and the return to pre-pandemic mobility patterns increased the demand for energy again. However, consumption remained below pre-pandemic levels as the effects of the pandemic continued to shape energy and economic activities. The most obvious negative consequence of the pandemic was the increase in energy consumption by EU households. Given these short-term trends, it is clear that in order to ensure the EU achieves its goals by 2030, changes in all indicators should be continuously monitored and assessed. The main motivation of our study was therefore to propose a methodological approach that could provide information to make such monitoring possible.

---

## [Editor Report · Decision Letter 2]

15 Jan 2024

Progress on SDG 7 achieved by EU countries in relation to the target year 2030: a multidimensional indicator analysis using dynamic relative taxonomy

PONE-D-23-21807R2

Dear Dr. Dehnel,

We’re pleased to inform you that your manuscript has been judged scientifically suitable for publication and will be formally accepted for publication once it meets all outstanding technical requirements.

Kind regards,

André Ramalho, PhD

Academic Editor

PLOS ONE

---

## [Editor Report · Acceptance letter]

5 Feb 2024

PONE-D-23-21807R2 

PLOS ONE

Dear Dr. Dehnel, 

I'm pleased to inform you that your manuscript has been deemed suitable for publication in PLOS ONE. Congratulations! Your manuscript is now being handed over to our production team.

Kind regards, 

on behalf of

Prof. Dr. André Ramalho 

Academic Editor

PLOS ONE